# Divergence at the Interpolation Threshold: Identifying, Interpreting & Ablating the Sources of a Deep Learning Puzzle

## Abstract

Machine learning models misbehave, often in unexpected ways. One prominent misbehavior is when the test loss diverges at the interpolation threshold, perhaps best known from its distinctive appearance in double descent. While considerable theoretical effort has gone into understanding generalization of overparameterized models, less effort has been made at understanding why the test loss misbehaves at the interpolation threshold. Moreover, analytically solvable models in this area employ a range of assumptions and use complex techniques from random matrix theory, statistical mechanics, and kernel methods, making it difficult to assess when and why test error might diverge. In this work, we analytically study the simplest supervised model - ordinary linear regression - and show intuitively and rigorously when and why a divergence occurs at the interpolation threshold using basic linear algebra. We identify three interpretable factors that, when all present, cause the divergence. We demonstrate on real data that linear models' test losses diverge at the interpolation threshold and that the divergence disappears when we ablate any one of the three identified factors. We then leverage one of the three factors to construct *adversarial training data* that increases the test error by 1-3 orders of magnitude without affecting the training error. We conclude with contributing fresh insights to recent discoveries regarding superposition and double descent in nonlinear models.

## 1 Introduction

Machine learning models, while incredibly powerful, can sometimes act unpredictably. One of the most intriguing behaviors is when the test loss suddenly diverges at the interpolation threshold, a point where the model perfectly fits the training data, leading to zero training error. This phenomenon is distinctly observed in the double descent curve Belkin et al. (2019). Although much theoretical groundwork has been laid to comprehend generalization of overparameterized models (Vallet, 1989; Krogh & Hertz, 1991; Geman et al., 1992; Krogh & Hertz, 1992; Opper, 1995; Duin, 2000; Spigler et al., 2018; Belkin et al., 2019; Bartlett et al., 2020; Belkin et al., 2020; Nakkiran et al., 2021; Poggio et al., 2019; Advani et al., 2020; Liang & Rakhlin, 2020; Adlam & Pennington, 2020; Rocks & Mehta, 2022b; 2021; 2022a; Mei & Montanari, 2022; Hastie et al., 2022; Bach, 2023), a general understanding of why test loss behaves erratically at this threshold remains elusive. Many analytical models aiming to explain this behavior rely on a plethora of assumptions (e.g., i.i.d additive Gaussian noise, sub-Gaussian covariates, $(8 + m)$-moments) and use advanced proof techniques from random matrix theory, statistical mechanics, and kernel methods. This complexity muddies the waters, making it challenging to pinpoint the general conditions leading to test error misbehavior. For instance, a recent study on toy nonlinear autoencoders by Anthropic unveiled a divergence even in the absence of noise (Henighan et al., 2023), an assumption that many theories relied upon (Bartlett et al., 2020; Liang & Rakhlin, 2020; Belkin et al., 2020; Hastie et al., 2022; Mei & Montanari, 2022; Bach, 2023). This unexpected outcome prompts the question: with all this theory, should we have expected the result?

In this work, we intuitively and quantitatively explain why the test loss diverges at the interpolation threshold, without assumptions and without resorting to intricate mathematical tools (e.g., random matrix theory, replica calculations, reproducing kernel Hilbert spaces) but also without sacrificing

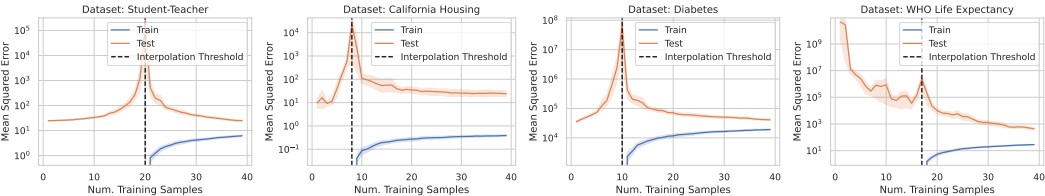

Figure 1: **Ordinary linear regression exhibits a divergence at the interpolation threshold on synthetic and real data.** Left to Right: Synthetic, California Housing (Pace & Barry, 1997), Diabetes (Efron et al., 2004), World Health Organization Life Expectancy (Gochiashvili, 2023). A divergence can occur as one approaches the interpolation threshold because (1) at least one singular mode in the training features becomes probabilistically likely to have a small amount of non-zero variance, (2) whose relation to the training regression targets is error-prone, such that when (3) new test data varies greatly along this direction, the model is forced to extrapolate distantly along a mode it doesn't understand well, causing the error to explode. Blue is training error, Orange is test error.

precision. To accomplish this, we focus on the simplest supervised model - ordinary linear regression - using the most basic linear algebra primitive: the singular value decomposition. We identify three distinct interpretable factors which, when collectively present, trigger the divergence. Through practical experiments on real data sets, we confirm that both model's test losses diverge at the interpolation threshold, and this divergence vanishes when even one of the three factors is removed. We then use one of the three factors to construct adversarial training data that We complement our understanding by offering a geometric picture that reveals linear models perform representation learning when overparameterized, and conclude by shedding light on recent results in nonlinear models concerning superposition (Henighan et al., 2023). By building general understanding, we offer valuable insights into surprising behaviors observed in nonlinear models.

## 2 DIVERGENCE IN ORDINARY LINEAR REGRESSION

To offer an intuitive yet quantitative understanding of model misbehavior, we turn to ordinary linear regression. Ordinary linear regression is useful for its simplicity, and because closed-form solutions are known for both the underparameterized and overparameterized regimes, meaning we can avoid complexity by excluding a learning algorithm and its corresponding learning dynamics.

**Notation and Terminology** Consider a regression dataset of $N$ training data with features $\vec{x}_n \in \mathbb{R}^D$ and targets $y_n \in \mathbb{R}$. We sometimes use matrix-vector notation to refer to the training data: $X \in \mathbb{R}^{N \times D}$ and $Y \in \mathbb{R}^{N \times 1}$. In ordinary linear regression, we want to learn parameters $\hat{\vec{\beta}} \in \mathbb{R}^D$ such that $\vec{x}_n \cdot \hat{\vec{\beta}} \approx y_n$. We will study three key parameters: number of model parameters $P$, number of training data $N$, and dimensionality of the data $D$. We say that a model is *overparameterized* (a.k.a. underconstrained) if $N < P$ and *underparameterized* (a.k.a. overconstrained) if $N > P$. The *interpolation threshold* refers to $N = P$, because when $N \leq P$, the model can perfectly interpolate the training points. Recall that in ordinary linear regression, the number of parameters $P$ equals the dimension $D$ of the covariates. Consequently, rather than thinking about changing the number of parameters $P$, we'll instead think about changing the number of data points $N$.

**Empirical Evidence on Synthetic & Real Data** Before studying ordinary linear regression mathematically, does our claim that it exhibits a divergence at the interpolation threshold hold empirically? We show that it indeed does, using one synthetic and three real datasets: World Health Organization Life Expectancy (Gochiashvili, 2023), California Housing (Pace & Barry, 1997), Diabetes (Efron et al., 2004); these three real datasets were selected on the basis of being easily accessible through sklearn (Pedregosa et al., 2011) or Kaggle. All display a spike in test mean squared error at the interpolation threshold (Fig. 1). Our code will be publicly available.

**Mathematical Analysis of Ordinary Linear Regression** To understand under what conditions and why the test loss diverges at the interpolation threshold in linear regression, we'll study the two

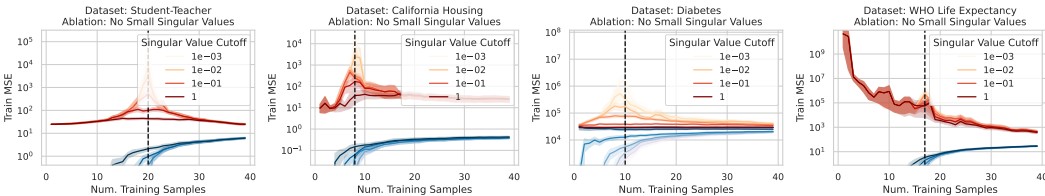

Figure 2: **Required Factor #1: How much training features vary in each direction.** The test loss diverges at the interpolation threshold only if training features $X$ contain small (non-zero) singular values. Ablation: By removing all singular values below a cutoff, the divergence at the interpolation threshold is diminished or disappears entirely. Blue is training error, Orange is test error.

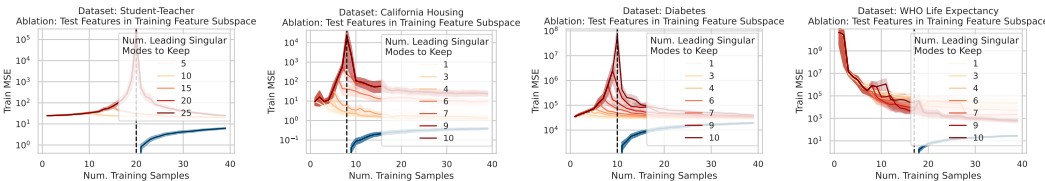

Figure 3: **Required Factor #2: How much, and in which directions, test features vary relative to training features.** The test loss diverges only if the test features $\vec{x}_{test}$ have a large projection onto the training features $X$'s right singular vectors $V$. Ablation: By projecting the test features into the subspace of the leading singular modes, the divergence at the interpolation threshold is diminished or disappears entirely. Blue is training error, Orange is test error.

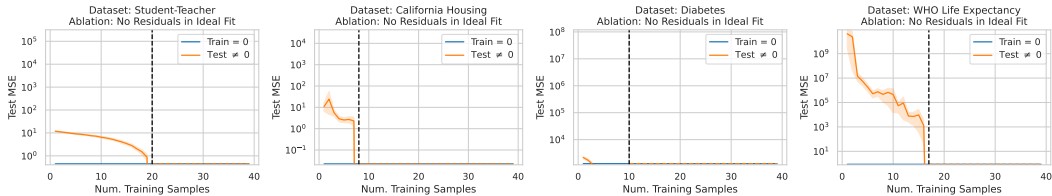

Figure 4: **Required Factor #3: How well the best possible model in the model class can correlate variance in training features with training targets.** The test loss diverges only if the residuals $E$ from the best possible model in the model class on the training data have a large projection onto the training features $X$'s left singular vectors $U$. Ablation: By ensuring the true relationship between features and targets is within the model class i.e. linear, the divergence at the interpolation threshold disappears. Blue is training error, Orange is test error.

parameterization regimes. If the regression is underparameterized, we estimate the linear relationship between covariates $\vec{x}_n$ and target $y_n$ by solving the least-squares minimization problem:

$$\hat{\vec{\beta}}_{under} \stackrel{\text{def}}{=} \arg\min_{\vec{\beta}} \frac{1}{N} \sum_n ||\vec{x}_n \cdot \vec{\beta} - y_n||_2^2 = \arg\min_{\vec{\beta}} ||X\vec{\beta} - Y||_2^2.$$

The solution is the ordinary least squares estimator based on the second moment matrix $X^T X$:

$$\hat{\vec{\beta}}_{under} = (X^T X)^{-1} X^T Y.$$

If the model is overparameterized, the optimization problem is ill-posed since we have fewer constraints than parameters. Consequently, we choose a different (constrained) optimization problem:

$$\hat{\vec{\beta}}_{over} \stackrel{\text{def}}{=} \arg\min_{\vec{\beta}} ||\vec{\beta}||_2^2 \qquad \text{s.t.} \qquad \forall\, n \in \{1, ..., N\} \quad \vec{x}_n \cdot \vec{\beta} = y_n.$$

We choose this optimization problem because it is the one gradient descent implicitly minimizes (App. B). The solution to this optimization problem uses the Gram matrix $X X^T \in \mathbb{R}^{N \times N}$:

$$\hat{\vec{\beta}}_{over} = X^T (X X^T)^{-1} Y.$$

One way to see why the Gram matrix appears is via constrained optimization: define the Lagrangian $\mathcal{L}(\vec{\beta}, \vec{\lambda}) \stackrel{\text{def}}{=} \frac{1}{2}||\vec{\beta}||_2^2 + \vec{\lambda}^T(Y - X\vec{\beta})$ with Lagrange multipliers $\vec{\lambda} \in \mathbb{R}^N$, then differentiate with respect to the parameters and Lagrange multipliers to obtain the overparameterized solution. After being fit, for test point $\vec{x}_{test}$, the model will make the following predictions:

$$\hat{y}_{test,under} = \vec{x}_{test} \cdot \hat{\vec{\beta}}_{under} = \vec{x}_{test} \cdot (X^T X)^{-1} X^T Y$$

$$\hat{y}_{test,over} = \vec{x}_{test} \cdot \hat{\vec{\beta}}_{over} = \vec{x}_{test} \cdot X^T (X X^T)^{-1} Y.$$

Hidden in the above equations is an interaction between three quantities that can, when all grow extreme, create a divergence in the test loss. To reveal the three quantities, we'll rewrite the regression targets by introducing a slightly more detailed notation. Unknown to us, there are some ideal linear parameters $\vec{\beta}^* \in \mathbb{R}^P = \mathbb{R}^D$ that truly minimize the test mean squared error. We can write any regression target as the inner product of the data $\vec{x}_n$ and the ideal parameters $\beta^*$, plus an additional error term $e_n$ that is an "uncapturable" residual from the "perspective" of the model class $y_n = \vec{x}_n \cdot \vec{\beta}^* + e_n$.. In matrix-vector form, we will equivalently write:

$$Y = X\vec{\beta}^* + E,$$

with $E \in \mathbb{R}^{N \times 1}$. To be clear, we are *not* imposing assumptions. Rather, we are introducing notation to express that there are (unknown) ideal linear parameters, and possibly non-zero errors $E$ that even the ideal model might be unable to capture; these errors $E$ could be random noise or could be fully deterministic patterns that this particular model class cannot capture. Using this new notation, we rewrite the model's predictions to show how the test datum's features $\vec{x}_{test}$, training data's features $X$ and training data's regression targets $Y$ interact. In the underparameterized regime:

$$\hat{y}_{test,under} = \vec{x}_{test} \cdot \beta^* + \vec{x}_{test} \cdot (X^T X)^{-1} X^T E$$

$$\hat{y}_{test,under} - y^*_{test} = \vec{x}_{test} \cdot (X^T X)^{-1} X^T E.$$

This equation is important, but opaque. To extract the intuition, let $y^*_{test} \stackrel{\text{def}}{=} \vec{x}_{test} \cdot \beta^*$ and replace $X$ with its singular value decomposition $X = U\Sigma V^T$. Let $R \stackrel{\text{def}}{=} \text{rank}(X)$ and let $\sigma_1 > \sigma_2 > ... > \sigma_R > 0$ be $X$'s (non-zero) singular values. We can decompose the underparameterized prediction error $\hat{y}_{test,under} - y^*_{test}$ along the orthogonal singular modes:

$$\hat{y}_{test,under} - y^*_{test} = \vec{x}_{test} \cdot V\Sigma^+ U^T E = \sum_{r=1}^{R} \frac{1}{\sigma_r} (\vec{x}_{test} \cdot \vec{v}_r)(\vec{u}_r \cdot E).$$

In the overparameterized regime, our calculations change slightly:

$$\hat{y}_{test,over} - y_{test}^* = \vec{x}_{test} \cdot (X^T(XX^T)^{-1}X - I_D)\beta^* + \vec{x}_{test} \cdot (X^TX)^{-1}X^TE.$$

If we again replace $X$ with its SVD $USV^T$, we can again simplify $\vec{x}_{test} \cdot (X^TX)^{-1}X^TE$. This yields our final equations for the prediction errors.

$$\hat{y}_{test,over} - y_{test}^* = \sum_{r=1}^{R} \frac{1}{\sigma_r}(\vec{x}_{test} \cdot \vec{v}_r)(\vec{u}_r \cdot E) + \vec{x}_{test} \cdot (X^T(XX^T)^{-1}X - I_D)\beta^*$$

$$\hat{y}_{test,under} - y_{test}^* = \sum_{r=1}^{R} \frac{1}{\sigma_r}(\vec{x}_{test} \cdot \vec{v}_r)(\vec{u}_r \cdot E).$$

The shared term between the two predictions causes the divergence:

$$\sum_{r=1}^{R} \frac{1}{\sigma_r}(\vec{x}_{test} \cdot \vec{v}_r)(\vec{u}_r \cdot E). \tag{1}$$

*Eqn. 1 is critical.* It reveals that our test prediction error (and thus, our test squared error!) will depend on an interaction between 3 quantities:

1. How much the *training features* $X$ vary in each direction (Fig. 2); more formally, the inverse (non-zero) singular values of the *training features* $X$:

$$\frac{1}{\sigma_r}.$$

2. How much, and in which directions, the *test features* $\vec{x}_{test}$ vary relative to the *training features* $X$ (Fig. 3); more formally: how $\vec{x}_{test}$ projects onto $X$'s right singular vectors $V$:

$$\vec{x}_{test} \cdot \vec{v}_r.$$

3. How well the *best possible model in the model class* can correlate the variance in the *training features* $X$ with the *training regression targets* $Y$ (Fig. 4); more formally: how the residuals $E$ of the best possible model in the model class (i.e. insurmountable "errors" from the "perspective" of the model class) project onto $X$'s left singular vectors $U$:

$$\vec{u}_r \cdot E.$$

When (1) and (3) co-occur, the model's parameters along this singular mode are likely incorrect. When (2) is added to the mix by a test datum $\vec{x}_{test}$ with a large projection along this mode, the model is forced to extrapolate significantly beyond what it saw in the training data, in a direction where the training data had an error-prone relationship between its predictions and the training targets, using parameters that are likely wrong. As a consequence, the test squared error explodes!

For completeness, recall the overparameterized prediction error $\hat{y}_{test,over} - y_{test}^*$ has another term:

$$\vec{x}_{test} \cdot (X^T(XX^T)^{-1}X - I_D)\beta^*. \tag{2}$$

To understand why this bias exists, recall that our goal is to correlate fluctuations in the covariates $\vec{x}$ with fluctuations in the targets $y$. In the overparameterized regime, there are more parameters than data; consequently, for $N$ data points in $D = P$ dimensions, the model can "see" fluctuations in at most $N$ dimensions, but has no "visibility" into the remaining $P - N$ dimensions. This causes information about the optimal linear relationship $\vec{\beta}^*$ to be lost, thereby increasing the overparameterized prediction error $\hat{y}_{test,over} - y_{test}^*$.

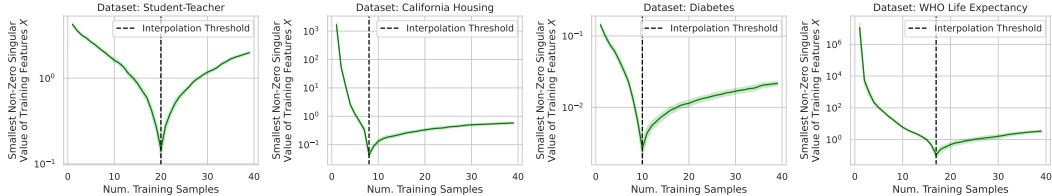

Figure 5: **The training features are most likely to obtain their smallest non-zero singular value when approaching the interpolation threshold.** This means that the first required factor for a divergence (small non-zero singular values; Fig. 2) is likely to occur near the interpolation threshold.

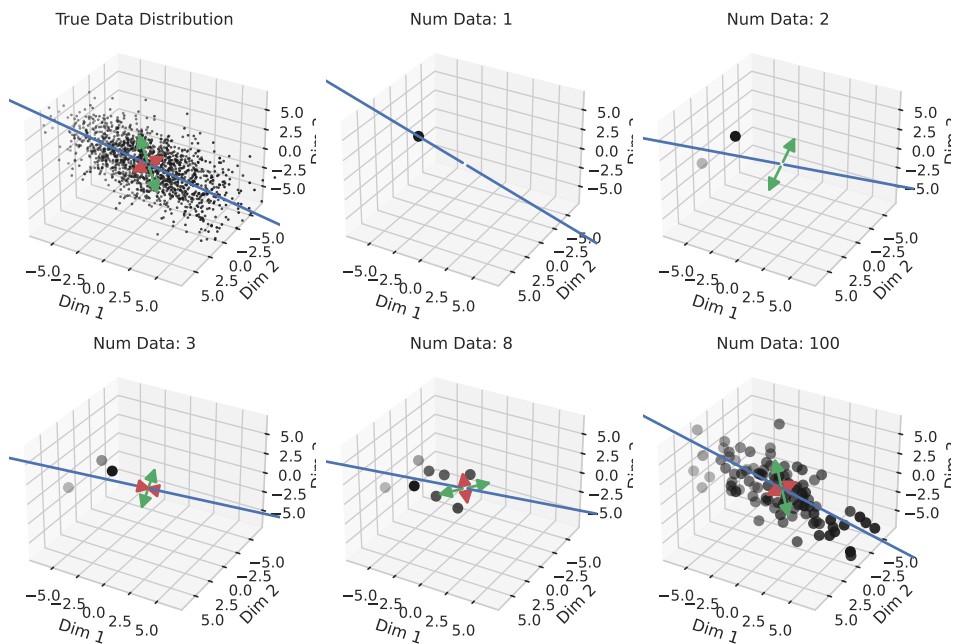

Figure 6: **Geometric intuition for why the smallest non-zero singular value reaches its lowest value near the interpolation threshold.** If 1 datum is observed, variance exists in only 1 direction. If 2 data are observed, a second axis of variation appears, but because the two data are likely to share some component, the second axis is likely to have less variation than the first. At the interpolation threshold (here, $D = P = N = 3$), because the three data are likely to share components along the first two axes, the third axis is likely to have even less variance. Beyond the interpolation threshold, additional data contribute additional variance to these three axes.

**Divergence at the Interpolation Threshold** Why does this divergence happen near the interpolation threshold? The answer is that the first factor (small non-zero singular values in the training features $X$) is likely to occur at the interpolation threshold (Fig. 5), but why? Suppose we're given a single training datum $\vec{x}_1$. So long as this datum isn't exactly zero, that datum varies in a single direction, meaning we gain information about the variance in that direction, but the variance in all orthogonal directions is exactly 0. With the second training datum $\vec{x}_2$, so long as this datum isn't exactly zero, that datum varies, but now, some fraction of $\vec{x}_2$ might have a positive projection along $\vec{x}_1$; if this happens (and it likely will, since the two vectors are unlikely to be exactly orthogonal), the shared direction gives us *more* information about the variance in this shared direction, but *less* information about the second orthogonal direction of variation. Ergo, the training data's smallest non-zero singular value after 2 samples is probabilistically smaller than after 1 sample. As we approach the interpolation threshold, the probability that each additional datum has large variance in a new direction orthogonal to all previous directions grows unlikely (Fig. 6), but as we move beyond the interpolation threshold, the variance in each covariate dimension becomes increasingly clear.

Figure 7: **Adversarial Training Datasets in Linear Regression.** By manipulating the residual errors $E$ that the best possible model in the model class achieves on the training data, we construct training datasets that increase the test error of the learned model by 1-3 orders of magnitude without affecting its training error. Blue is training error, Orange is test error.

**Ablating the Divergence** The test loss will not diverge if any of the three required factors are absent. What could cause that?

- *Small-but-nonzero singular values do not appear in the training data features.* One way to accomplish this is by setting all singular values below a selected threshold to exactly $0$.

- *The test datum does not vary in different directions than the training features.* If the test datum lies entirely in the subspace of just a few of the leading singular directions, then the divergence is unlikely to occur.

- *The best possible model in the model class makes no errors on the training data.* For example, if we use a linear model class on data where the true relationship is a noiseless linear one, then at the interpolation threshold, we will have $D = P$ data, $P = D$ parameters, our line of best fit will exactly match the true relationship, and no divergence will occur.

To test our understanding, we independently ablate each factor:

1. No Small Singular Values in Training Features: As we run the ordinary linear regression fitting process, and as we sweep the number of training data, we also sweep different singular value cutoffs and remove all singular values of the training features $X$ below the cutoff.

2. Test Features Lie in the Training Features Subspace: As we run the ordinary linear regression fitting process, as we sweep the number of training data, we project the test features $\vec{x}_{test}$ onto the subspace spanned by the training features $X$ singular modes.

3. No Residual Errors in the Optimal Model: We first use the entire dataset to fit a linear model $\vec{\beta}^*$, then replace $Y$ with $X\vec{\beta}^*$ and $y_{test}^*$ with $\vec{x}_{test} \cdot \vec{\beta}^*$ to ensure the true relationship is linear. We then rerun our typical fitting process, sweeping the number of training data.

We intentionally apply each ablation individually to our one synthetic dataset and three real datasets, and find that each ablation partially or wholly prevents a divergence from occurring (Figs. 2 3 4).

## 3 CONSTRUCTING ADVERSARIAL TRAINING DATA

Adversarial test examples are well known Szegedy et al. (2013); Goodfellow et al. (2014); Kurakin et al. (2018); Athalye et al. (2018); Xie et al. (2022), and are visible in our analysis as Factor 2 i.e. how much, and in which direction, the test features $\vec{x}_{test}$ vary relative to the training features $X$ (App. Fig. 11). More interestingly, Factor 3 in our analysis reveals the existence of *adversarial training data*. Recalling that Factor 3 is how well *the best possible model in the model class* can correlate the variance in the training features $X$ with the training regression targets $Y$, Factor 3 tells us that by amplifying the training residual errors $E$ along the smallest singular mode, one can significantly increase the test error without affecting the training error. We then empirically test and confirm this insight (Fig. 7). Adversarial training data is akin to targeted dataset poisoning attacks Biggio et al. (2012); Steinhardt et al. (2017); Wallace et al. (2020); Carlini & Terzis (2021); Carlini (2021); Schuster et al. (2021) or backdoor attacks Chen et al. (2017); Gu et al. (2017); Carlini & Terzis (2021), but differs in that we manipulate the model to misbehave on *all* test data rather than misbehave on *specific* test data. The key insight is that the test error of the *learned model* is increased due to errors that the *best possible model in the model class* makes on the training data.

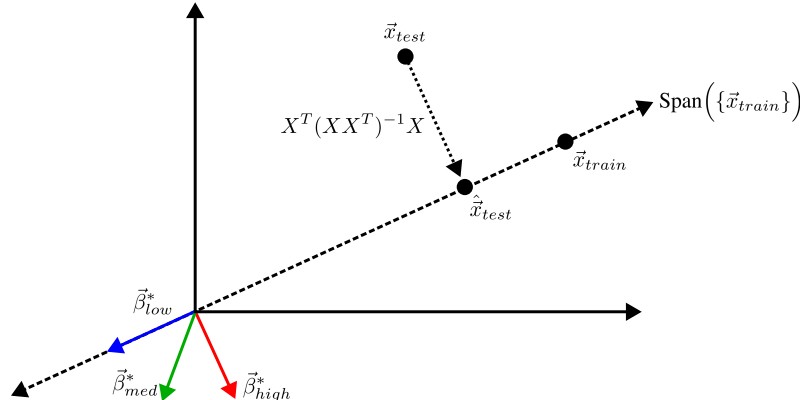

Figure 8: **Geometry of Generalization in Overparameterized Ordinary Linear Regression.** The rowspace of the training features $X$ forms a subspace (here, $\mathbb{R}^1$) of the ambient space (here, $\mathbb{R}^2$). For test datum $\vec{x}_{test}$, the linear model forms an internal representation of the test datum $\hat{\vec{x}}_{test}$ by orthogonally projecting the test datum onto the rowspace via projection matrix $X^T(XX^T)^{-1}X$. The generalization error will then increase commensurate with the inner product between $\hat{\vec{x}}_{test} - \vec{x}_{test}$ and the best possible parameters for the function class $\vec{\beta}^*$. Three different possible $\vec{\beta}^*$ are shown with low, medium and high generalization errors.

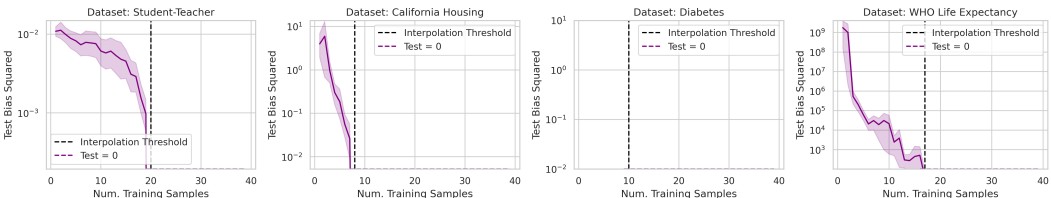

Figure 9: **Test Error of Overparameterized Models.** Large inner product between the ideal model's parameters and the difference between the fit model's internal representations of the test data and the test data creates large test error.

## 4 GENERALIZATION IN OVERPARAMETERIZED LINEAR REGRESSION

A natural question to ask is under what conditions will an overparameterized model generalize well? We previously saw that away from the interpolation threshold, the variance (Eqn. 1) is unlikely to affect the discrepancy between the overparameterized model's predictions and the ideal model's predictions, meaning most of the discrepancy must therefore emerge from the bias (Eqn. 2). This bias term yields an intuitive geometric picture (Fig. 8) that also reveals a surprising fact: *overparameterized linear regression does representation learning!* Specifically, for test datum $\vec{x}_{test}$, a linear model creates a representation of the test datum $\hat{\vec{x}}_{test}$ by orthogonally projecting the test datum onto the row space of the training covariates $X$ via the projection matrix $X^T(XX^T)^{-1}X$:

$$\hat{\vec{x}}_{test} \stackrel{\text{def}}{=} X^T(XX^T)^{-1}X \, \vec{x}_{test}.$$

Seen this way, the bias can be rewritten as the inner product between (1) the difference between its representation of the test datum and the test datum and (2) the ideal linear model's fit parameters:

$$(\hat{\vec{x}}_{test} - \vec{x}_{test}) \cdot \vec{\beta}^*. \tag{3}$$

Intuitively, an overparameterized model will generalize well if the model's representations capture the essential information necessary for the best model in the model class to perform well (Fig. 9).

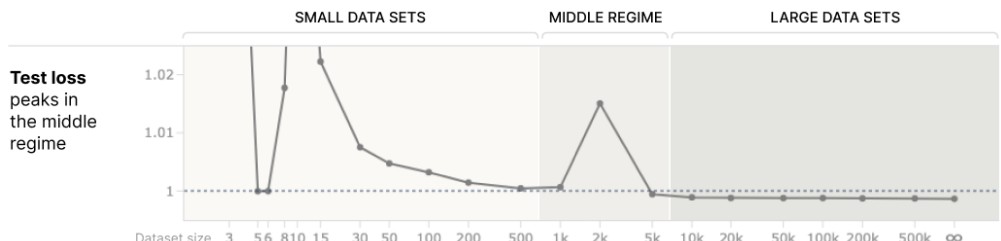

Figure 10: Henighan et al. (2023) recently discovered a divergence in test loss at the interpolation threshold in noiseless nonlinear autoencoders. Despite significant theoretical work in this area, whether such a phenomenon should be expected is unclear because of the numerous assumptions and particular model classes studied by prior work.

## 5 INTUITION EXTENDS TO NONLINEAR MODELS

Although we mathematically studied ordinary linear regression, the intuition for why the test loss diverges extends to nonlinear models, such as polynomial regression and including certain classes of deep neural networks (Jacot et al., 2018; Lee et al., 2017; Bordelon et al., 2020). For a concrete example about how our intuition can shed light on the behavior of nonlinear models, Henighan et al. (2023) recently discovered interesting properties of shallow nonlinear autoencoders: depending on the number of training data, (1) autoencoders either store data points or features, and (2) the test loss increases sharply between these two regimes (Fig. 10). Our work sheds light on the results in two ways:

1. Henighan et al. (2023) write, "It's interesting to note that we're observing double descent in the absence of label noise." Our work clarifies that noise, in the sense of a random quantity, is *not* necessary to produce double descent. Rather, what is necessary is *residual errors from the perspective of the model class - $E$*, in our notation. Those errors could be entirely deterministic, such as a nonlinear model attempting to fit a noiseless linear relationship, or other model misspecifications.

2. Henighan et al. (2023) write, "[Our work] suggests a naive mechanistic theory of overfitting and memorization: memorization and overfitting occur when models operate on 'data point features' instead of 'generalizing features'." Our work hopefully clarifies that this terminology can be made more precise: when overparameterized, "data point features" are akin to the Gram matrix $XX^T$ and when underparameterized, "generalizing features" are akin to the second moment matrix $X^TX$. Our work hopefully clarifies that "data point features" can and very often do generalize, and that there is a deep connection between the two, i.e., their shared spectra.

## 6 DISCUSSION

In this work, we intuitively and quantitatively explained why the test loss misbehaves based on three interpretable factors, tested our understanding via ablations, and added conceptual clarity of recent discoveries in nonlinear models (Henighan et al., 2023).

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

Figure 11: **Adversarial Test Examples in Linear Regression.** Adversarial examples arise by pushing $\vec{x}_{test}$ far along the trailing singular modes in the training features $X$. Blue is training error, Orange is test error.

## A  ADVERSARIAL TEST EXAMPLES IN LINEAR REGRESSION

Factor 2 in our analysis corresponds to advesarial test examples. Recall that Factor 2 is how much, and in which directions, the *test features* $\vec{x}_{test}$ vary relative to the *training features* $X$ (Fig. 2), or moremore formally: how $\vec{x}_{test}$ projects onto $X$'s right singular vectors $V$:

$$\vec{x}_{test} \cdot \vec{v}_r.$$

We visually display such adversarial attacks (Fig. 11).

## B  WHY GRADIENT DESCENT IMPLICITLY REGULARIZES

This is a sketch of why gradient descent implicitly regularizes. Suppose we have a model $Xw$ for a vector of data $y \in \mathbb{R}^n$ and want to minimize the norm of the error,

$$L(w) = \|Xw - y\|_2^2 = \|e\|_2^2$$

where we introduce some short-hand notation. We use the gradient learning rule,

$$w(t+1) = w(t) - \eta X^T e(t)$$

$$\Rightarrow e(t+1) = e(t) - \eta X X^T e(t)$$

$$\Rightarrow e(t+1) = (I - \eta X X^T)e(t)$$

Each matrix satisfies $X \in \mathbb{R}^{n \times d_1}$ where $n$ is the number of samples and $d_1$ is the dimension of each sample. In the overparameterized setting we have $d_1 > n$ and so $XX^T$ will generically have full-rank and the error will go to zero.

This lies in the difference between $XX^T$ which appears here in the error analysis and $X^T X$ which appears in the solution. So we can have $XX^T \in \mathbb{R}^{n \times n}$ generically full-rank only if we have more parameters than there is data. On the other hand, we only have $X^T X$ full-rank if also it's satisfied that there is more data than parameters. This is important because in this case we can compute the pseudo-inverse easily. Generically, we can show that if we use gradient descent we have something like the following,

$$\underbrace{(X^T X)^{-1} X}_{\text{left inverse}} \quad \underbrace{X^{-1}}_{\text{inverse}} \quad \underbrace{X^T (XX^T)^{-1}}_{\text{right inverse}}$$

for the cases where we are under-parameterized, minimally parameterized, or over-parameterized to model the data.

So under gradient flow we'll suppose the parameters update according to,

$$\dot{w} = -\eta X^T e$$

$$w(0) = 0$$

Observe that the gradient $\dot{w}$ is invariantly in the span of $X^T$ so we may conclude that $w(t)$ is always in the span of $X^T$. Generically, any solution in the over-parameterized setting is a global optimizer

such that $Xw = y$. This means that the limit of the flow can be written as $w^* = X^T \alpha$ for some coefficient vector with the constraint that $Xw^* = y$. After some manipulations we find that,

$$y = Xw^* = XX^T \alpha$$
$$\Rightarrow \alpha = (XX^T)^{-1} y$$
$$\Rightarrow w^* = X^T (XX^T)^{-1} y = X^+ y$$

This means that the solution $X^+$ picked from gradient flow is the Moore-Penrose psuedoinverse. This can be defined as the matrix,

$$X^+ = \lim_{\lambda \to 0^+} X^T (XX^T + \lambda I)^{-1}$$

Also observe that there is a unique minimizer for the regularized problem,

$$\min_w L(w) + \lambda \|w\|_2^2$$

with value $w_\lambda = X^T (XX^T + \lambda I)^{-1} y$. Perhaps, $Xw = y$ has a set of solutions, but it is clear this set is convex so there is a unique minimum norm solution. On the other hand, each $w_\lambda$ corresponds to a best solution with norm below the minimum. However, we have $w^* = \lim_{\lambda \to 0^+} w_\lambda$ from continuity. Since $w^*$ is an exact solution it can't have less than the minimum-norm and it is clear $w^*$ can't have above the minimum-norm either since this is not the case for any of the $w_\lambda$. We conclude that gradient descent does indeed find the minimum norm solution.

