# OpenReview forum: "Divergence at the Interpolation Threshold: Identifying, Interpreting & Ablating the Sources of a Deep Learning Puzzle"
_ICLR.cc/2024/Conference — ICLR 2024 Conference Withdrawn Submission_

### Official Review · Reviewer_5RFc · 2023-10-17

**Soundness:** 4 excellent
**Presentation:** 2 fair
**Contribution:** 4 excellent
**Rating:** 6
**Confidence:** 4

**Summary:**

This work investigates the nature of the divergence of test performance and the interpolation threshold of double descent for ordinary least squares. The authors use simple linear algebra to decompose the loss into three terms (and a bias in the overparameterized case) that can be easily interpreted as factors that contribute to this phenomenon. Experiments are performed that ablate these three factors verifying that they provide a useful perspective. Next, the authors illustrate how the bias term of overparameterized regression can be interpreted as capturing the interaction between the size of the projection of the test point into the training subspace and the ideal parameters which introduces a representation learning perspective. Finally, they discuss how these intuitions might extend beyond the linear setting.

**Strengths:**

* I found the decomposition of the error in eqns (1) and (2) to provide a very clear perspective on the cause of divergence which, to the best of my knowledge, has not been previously derived.
* I appreciated the minimalist approach to this paper. I think the core message is sufficient to be worthy of publication and the paper benefits from not being distracted by additional points.
* I appreciated the intuition of why it is increasingly likely that the Nth vector is unlikely to vary in the new dimension it has introduced. This was a nice perspective. It might even be nice to drive this home by providing a concrete example with, for example, Gaussian vectors where you could prove that each new singular value introduced is increasingly likely to be small (but this is at the discretion of the authors, not something I think is required).
* I enjoyed the representation learning perspective of the bias term on p8 and in Fig 8. I found this perspective to be helpful in the context of this work.
* I thought the experiments were sufficiently well constructed to illustrate the points made in the text (although I do have some concerns about the presentation of results included in the weaknesses section).

**Weaknesses:**

Overall, I am very positive about this work and think its core points (i.e. the decomposition and the projection of a test point) are instructive and a useful contribution. However, especially given that the goal of this paper was to be simple and accessible, I think that improving the exposition would greatly improve this work. I include some suggested improvements below and would be open to increasing my score if a revised version addressed these points.

* I think the statement on page 1 that "our goal is to explain intuitively and quantitatively why the test loss diverges at the interpolation threshold, without assumptions and without resorting to intricate mathematical tool" might be a very slight misrepresentation without mentioning that this is restricted to linear models.
* I thought Fig 1 to be a strange choice and would prefer to see something that illustrates the message of the paper. Why not provide an illustrative plot of double descent in test loss with the decomposition of the three factors plotted in parallel?
* The other figures could also be improved in terms of clarity. The text size is far too small. The legends are sometimes incomplete (e.g. Fig 3 + 4 - what are the blue lines?). The titles are unclear (what does "condition" mean?)
* I thought Fig 7 in particular could do with some work. It is currently very difficult to understand exactly what is happening in each panel.
* Given that the 3 bullet points on pages 5 & 6 are arguably the main point of the paper, I think their explanations could be sharpened somewhat. Particularly for the second point, I didn't find the text description to be intuitive and resorted to interpreting the expression myself. This is of course just a readability suggestion. It would also be nice to explicitly link each point to its corresponding plot.
* There seems to be some contradiction in the ablation discussion on p7+8. It suggests "switching from ordinary linear regression to ridge regression", however, the actual ablation implemented is to "sweep different singular value cutoffs" and remove corresponding values. These are different things so it's not clear why ridge regression, which would apply shrinkage instead, is implied.
* Similarly to previous figures, I thought the clarity of Fig 8 could be improved. In particular, the final two sentences of the caption could be spelled out better in the figure (and maybe also the caption). I don't think the current illustration does this final step justice (which is much more clear in the main text of the paper). I would suggest looking at e.g. Fig 3.4 of [1] for inspiration on how to improve the clarity of this style of figure.
* It would be nice to include the code used to produce the results in this paper such that readers can easily interact with the experiments.
* I think this work could have done a better job of contextualizing itself relative to other similar works (e.g. [2]). It would be nice to explain what other similar analyses of double descent (at least in the linear setting) have discovered about this phenomenon.


Minor:
* "How much the training features X vary in each direction" - It might be clearer to specify that you refer to a direction r in the SVD rather than in the original X space.
* page 5, point 2 references the wrong figure.


[1] Hastie, T., Tibshirani, R., Friedman, J. H., & Friedman, J. H. (2009). The elements of statistical learning: data mining, inference, and prediction (Vol. 2, pp. 1-758). New York: springer.

[2] Hastie, T., Montanari, A., Rosset, S., & Tibshirani, R. J. (2022). Surprises in high-dimensional ridgeless least squares interpolation. Annals of statistics, 50(2), 949.

**Questions:**

* "recall that our goal is to correlate fluctuations in the covariates x with fluctuations in the targets y" - this line was a little unclear to me. Could the authors expand on what this means exactly? I would interpret this term as capturing the information/signal of the first N of the D total features and therefore misses whatever information/signal is included in those final D - N features. Are you saying something more than this here?

* I thought ablation 3 (obtaining $\beta^*$ via fitting to the full dataset) was a clever solution. However, I found Fig 5 difficult to follow. I think it would make more sense to illustrate the value of the $u_r \cdot E$ term (capturing the impact of the misspecification) so we can understand its contribution relative to the others. It's not totally clear to me how this would be combined with the current plot but at least the clarity of the current version should be improved.

---

> ### Author Response · Authors · 2023-11-21
> **Response to Reviewer 5RFc**
>
> We appreciate your feedback on our paper. Concrete, actionable feedback on how to improve is extremely valuable.
>
> Below is an incomplete response that we will complete tomorrow.
>
> > I think the statement on page 1 that "our goal is to [...] mathematical tool" might be a very slight misrepresentation without mentioning that this is restricted to linear models.
>
> Agreed. We will explicitly state the paper considers only linear models.
>
> > I thought Fig 1 to be a strange choice and would prefer to see something that illustrates the message of the paper.
>
> Agreed. We moved the (previous) Fig 1 to Section 5.
>
> > Why not provide an illustrative plot of double descent in test loss with the decomposition of the three factors plotted in parallel?
>
> We like this suggestion and tried reorganizing the figures to more clearly communicate this (see Page 3). Tomorrow, we’ll try creating a new figure that plots the 3 factors separately from their ablations.
>
> > The other figures could also be improved in terms of clarity. The text size is far too small. The legends are sometimes incomplete (e.g. Fig 3 + 4 - what are the blue lines?). The titles are unclear (what does "condition" mean?)
>
> Text size has been increased, legends made more clear, and titles edited. “Condition” meant “Ablation”; this should now be more clear.
>
> > I thought Fig 7 in particular could do with some work. It is currently very difficult to understand exactly what is happening in each panel.
>
> The overall exposition has been tightened up. We have expanded the explanation of the previous Fig 7 (now Fig 6) in the figure caption.
>
> > It would also be nice to explicitly link each point to its corresponding plot.
>
> Thank you - fixed!
>
> > There seems to be some contradiction in the ablation discussion on p7+8. [...] however, the actual ablation implemented is to "sweep different singular value cutoffs" and remove corresponding values. These are different things
>
> Your description is correct. We have fixed the incorrect text.
>
> > It would be nice to include the code used to produce the results in this paper such that readers can easily interact with the experiments.
>
> Code will be made publicly available. In the interim, we’ll share code with you tomorrow.
>
> > page 5, point 2 references the wrong figure.
>
> Thank you - fixed!

---

> ### Comment · Reviewer_5RFc · 2023-11-21
>
> I thank the authors for their partial response and I look forward to reading the complete response (and updated manuscript) once posted.

---

> > ### Author Response · Authors · 2023-11-22
> > **Response to Reviewer 5RFc**
> >
> > Based on our reviews, this paper will likely be rejected and it doesn't make sense to invest additional time (or ask more of yours). We think it makes more sense to submit to the ICLR 2024 blog track https://iclr-blogposts.github.io/2024/about/.
> >
> > That said, we appreciate your feedback tremendously and will continue to work to integrate it. If you're interested in helping us submit to the blog track, you are more than welcome to reach out :)

---

### Official Review · Reviewer_iMad · 2023-10-27

**Soundness:** 3 good
**Presentation:** 3 good
**Contribution:** 1 poor
**Rating:** 3
**Confidence:** 3

**Summary:**

This work investigates the least squares regression problem and identifies necessary conditions based on the SVD of the features for the test error to diverge at the interpolation threshold.

**Strengths:**

The paper is well written and easy to read. The intuitive discussion on the origin of the interpolation peak for least squares is interesting. The numerical experiments with real data sets illustrating the phenomenology and the visual schemes (e.g. Fig. 7 & 8) are helpful.

**Weaknesses:**

1. The result is very specific to least squares regression. For instance, the connection between the interpolation threshold and the rank of the features is a specificity of linear regression. Even in slightly more general settings, such as generalised linear models, this is not generally case. For example, in logistic regression the interpolation threshold corresponds the linear separability of the data [Rosset et al. 2003], and even for features that display a divergence at interpolation for regression display a kink rather than a peak for classification, see [Gerace et al. 2020] for an example in the context of random features. Although the authors argue that their criteria apply to other models (e.g. shallow auto-encoders), it is really not clear how general this is. The authors should consider adding at least a honest discussion of the limitations of this result.

2. The main results are contained in previous literature. For example: For instance, the relationship between the rank of the features and the interpolation threshold in linear regression has been discussed in [Hastie et al. 2022, Loureiro et al 2021], and the interplay between the features singular vectors and the target weights was studied extensively in [Wu & Xu 2020]. The fact that least-squares learns a representation which is the projection in the row-space of the features is a classical discussion in ML books [Bishop 2007]. The fact that double descent can occur with zero label noise due to model misspecification appeared in [Mei & Montanari 2022; Gerace et al. 2020; Loureiro et al 2021]. I appreciate the authors intention summarise and provide their own point of view to these results. However, I also feel that their justification that these previous works are "complex", "difficult" and "muddies the water" is subjective and comes across borderline disrespectful.

**Questions:**

- Although "under/overparameterised" is often used in the context of least-squares to refer to $N>D$ or $N<D$, this terminology is misleading, since in least-squares increasing the dimension $D$ both increases the number of parameters and decreases the sample complexity "N/D".

**Minor comments**
- The figures are small and hard to read. Different from Figs. 2-9, Fig. 1 is not in a vector format, so it gets pixelised when zoomed.

- To my best knowledge, the first works discussing the interpolation peak and how to mitigate them were [Opper et al. 1990; Krogh & Hertz (1991, 1992); Geman et al. 1992].

**References:**

[Rosset et al. 2003] Saharon Rosset, Ji Zhu, and Trevor J Hastie. Margin Maximizing Loss Functions.  Part of Advances in Neural Information Processing Systems 16 (NIPS 2003).

[Gerace et al. 2020] Federica Gerace, Bruno Loureiro, Florent Krzakala, Marc Mezard, Lenka Zdeborova. Generalisation error in learning with random features and the hidden manifold model. Proceedings of the 37th International Conference on Machine Learning, PMLR 119:3452-3462, 2020.

[Hastie et al. 2022]. Trevor Hastie, Andrea Montanari, Saharon Rosset, Ryan J. Tibshirani. Surprises in high-dimensional ridgeless least squares interpolation.  Ann. Statist. 50(2): 949-986 (April 2022). DOI: 10.1214/21-AOS2133

[Loureiro et al 2021]. Bruno Loureiro, Cedric Gerbelot, Hugo Cui, Sebastian Goldt, Florent Krzakala, Marc Mezard, Lenka Zdeborová.
Learning curves of generic features maps for realistic datasets with a teacher-student model. Part of Advances in Neural Information Processing Systems 34 (NeurIPS 2021).

[Wu & Xu 2020]. Denny Wu, Ji Xu. On the Optimal Weighted $\ell_2$ Regularization in Overparameterized Linear Regression. Part of Advances in Neural Information Processing Systems 33 (NeurIPS 2020).

[Mei & Montanari 2022] Song Mei and Andrea Montanari. The generalization error of random features regression: Precise asymptotics and the double descent curve. Communications on Pure and Applied Mathematics, 75(4):667–766, 2022.

[Bishop 2007]. Christopher M. Bishop. Pattern Recognition and Machine Learning. Springer, 2007.

[Opper et al. 1990] M Opper, W Kinzel, J Kleinz and R Nehl, *"On the ability of the optimal perceptron to generalise"*, 1990 J. Phys. A: Math. Gen. 23 L581.

[Krogh & Hertz 1991] A Krogh, J Hertz, *"A Simple Weight Decay Can Improve Generalization"*, NeurIPS 1991

[Krogh & Hertz 1992] A Krogh and J Hertz, *"Generalization in a linear perceptron in the presence of noise"*, 1992 J. Phys. A: Math. Gen. 25 1135.

[Geman et al. 1992] Geman, S., Bienenstock, E., and Doursat, R. Neural net-works and the bias/variance dilemma. Neural computation, 4(1):1–58, 1992.

---

> ### Author Response · Authors · 2023-11-21
> **Response to Reviewer iMad (1/2)**
>
> We appreciate your feedback on our paper. In response:
>
> > The main results are contained in previous literature [...] I appreciate the authors intention summarise and provide their own point of view to these results. However, I also feel that their justification that these previous works are "complex", "difficult" and "muddies the water" is subjective and comes across borderline disrespectful.
>
> We agree that the main results are contained in previous literature through one lens or another, and we have deep appreciation for the prior work.
>
> However, we believe that “complex” and “difficult” are objectively true. Our paper requires only two mathematical primitives: SVD and matrix calculus. We know of no other paper this simple.
>
> In contrast, consider the techniques used by some previous papers:
> - Wu & Xu 2020: Bounded 12th absolute central moment, asymptotic prediction risk, Stieltjes transform of limiting distribution described by Marchenko–Pastur distribution, self-consistent equations
> - Mei & Montanari 2022: random features models, proportional asymptotics, Gaussian processes, weak differentiability, convergence in probability, random matrix theory including Marchenko-Pastur distribution, Stieltjes transform
> - Hastie et al. 2022: Proportional asymptotics, uniformly bounded moments, sequences of deterministic PD matrices, random matrix theory, concentration of measure
>
> These techniques are harder than SVD + vector derivatives. We mean no disrespect by "complex" or "difficult". Our goal in writing this manuscript is to explain this phenomenon as simply as possible, to complement the precise-but-narrow results of previous papers with an imprecise-but-more-general result.
>
> Regarding “muddies the water,” we would be happy to use alternative phrasing. Our point - which we stand by - is that the numerous, specific and differing assumptions used by prior papers makes it difficult to determine which properties are necessary and/or sufficient for the phenomenon of interest. For example, consider the sentence:
>
> > The fact that double descent can occur with zero label noise due to model misspecification appeared in [Mei & Montanari 2022
>
> While Mei & Monanari 2022 did study model misspecification, they make multiple assumptions: covariates are uniform on the hypersphere, y has i.i.d. additive noise with finite 4th moment, the model is ridge regression, the learning algorithm is gradient flow, the solution requires large $N, n, d$, the nonlinear model component is a centered isotropic GP, the nonlinearity must be weakly differentiable, etc. Theorem 1’s result is that the prediction risk converges in probability to $(F_1^2 + \tau^2) \mathscr{R}(\rho, \zeta, \phi_1, \phi_2, \lambda/\mu_*^2)$. It is not obvious what qualitative behavior to generally expect, nor is it obvious how results change if assumptions change. Moreover, it is not obvious how the results depend on the _particular_ model misspecification considered versus general model misspecification.
>
> We adopted a methodology analogous to the model-experimental systems approach, prevalent in the natural sciences, in which simple and controllable toy models are studied. The goal is to develop an intuitive yet quantitative understanding of deep learning phenomena. Multiple published papers have built and non-rigorously analyze models of deep learning phenomena, with a prominent recent example of grokking:
>
> [1] Power et al. Grokking: Generalization Beyond Overfitting on Small Algorithmic Datasets.
> [2] Liu et al. Towards Understanding Grokking: An Effective Theory of Representation Learning.
> [3] Thilak et al. The Slingshot Mechanism: An Empirical Study of Adaptive Optimizers and the Grokking Phenomenon.
> [4] Nanda et al. Progress measures for grokking via mechanistic interpretability.
>
>
> > this terminology is misleading, since in least-squares increasing the dimension $D$ both increases the number of parameters and decreases the sample complexity "N/D".
>
> We conceptually think of fixed $D$: “Consequently, rather than thinking about changing the number of parameters P , we’ll instead think about changing the number of data points N.” We considered using the terminology “under/overconstrained” but decided this terminology was less common.
>
> > The figures are small and hard to read. Different from Figs. 2-9, Fig. 1 is not in a vector format, so it gets pixelised when zoomed.
>
> We converted figures from PNGs to PDFs to improve readability. Figure 1 is not ours, but we moved it to a less prominent position. We are limited by space in making the figures larger.
>
> > To my best knowledge, the first works discussing the interpolation peak and how to mitigate them were [Opper et al. 1990; Krogh & Hertz (1991, 1992); Geman et al. 1992].
>
> The earliest reference we found was Vallet 1989’s “The Hebb rule for learning linearly separable boolean functions: learning and generalization,” which we of course cited. We appreciate and have added these additional citations.

---

> ### Author Response · Authors · 2023-11-21
> **Response to Reviewer iMad (2/2)**
>
> > The result is very specific to least squares regression.
>
> This is true, but our goal was not to explain everything, nor was this our claimed contribution. Rather, our claimed contribution is to study the simplest model possible that exhibits this phenomenon of interest, and explain what causes it and why in quantitative terms with as much generality and intuition as possible.
>
> In other words, our contribution is not new conclusions, but an identification of the three general factors that are necessary to produce the divergence, along with a quantitative explanation of (a) why they are the correct factors to consider and (b) how they together interact to produce the divergence.
>
> If you know of prior work identifying these three factors and explaining why they generally matter, please do let us know!

---

> ### Author Response · Authors · 2023-11-22
> **Response to Reviewer iMad**
>
> Based on our reviews, this paper will likely be rejected and it doesn't make sense to invest additional time (or ask more of yours). We think it makes more sense to submit to the ICLR 2024 blog track https://iclr-blogposts.github.io/2024/about/.
>
> Thank you for your feedback and time.

---

### Official Review · Reviewer_7Ltc · 2023-11-07

**Soundness:** 3 good
**Presentation:** 3 good
**Contribution:** 1 poor
**Rating:** 3
**Confidence:** 3

**Summary:**

The paper investigates the double descent phenomenon within linear regression, particularly focusing on overparameterized least squares scenarios. It presents heuristic arguments and simple mathematical formulations to explain the occurrence of double descent. The authors dissect the prediction error into bias and variance components, further breaking down variance into three factors: the inverse of singular values, alignment with training sample directions, and model class limitations. While exploring how these factors contribute to double descent, the paper highlights the need for all three to be significant for the phenomenon to arise, noting the presence of bias in overparameterized settings and its potential impact on training loss.

**Strengths:**

1. The paper is well written and easy to follow.
2. The math is simple and results are all intuitive.

**Weaknesses:**

1. The paper's main contribution appears to rearticulate established "double descent" phenomena using simpler language, but it's unclear how this advances understanding beyond existing literature. Notably, there is an extensive body of work on double descent in the context of Kernel Ridge Regression (you can think of linear regression as an special case), such as the detailed discussion in Montanari et al. (https://arxiv.org/pdf/2308.13431.pdf), and references there.
2.  Decomposing the risk to examine the influence of individual components is a natural thing to do. The practicality of quantifying these terms in real-world applications remains questionable in your work. The paper does not seem to provide a more insightful analysis of risk behavior compared to prior studies that make assumptions on data to yield meaningful interpretations. It is essential for the paper to clarify how its approach contributes to the existing knowledge base in a way that is both significant and applicable.

I think this analysis is a nice way of thinking about double descent, like for a blog post or a lecture note, but I do not see any significant contribution.

**Questions:**

See weaknesses

---

> ### Author Response · Authors · 2023-11-19
> **Response to Reviewer 7Ltc**
>
> Thank you for your valuable feedback and insights on our manuscript. We appreciate the time and effort you've dedicated to reviewing our work. We would like to address the concerns raised and provide additional context to clarify our contributions to the field.
>
> >  it's unclear how this advances understanding beyond existing literature. Notably, there is an extensive body of work on double descent in the context of Kernel Ridge Regression (you can think of linear regression as an special case), such as the detailed discussion in Montanari et al. (https://arxiv.org/pdf/2308.13431.pdf), and references there.
>
> Our work was motivated by a conversation with Montanari himself. We had a different research problem, related to whether one should expect a particular overparameterized model to generalize in a particular setting, and met with Montanari who kindly walked us through different results for multiple analytically solvable models. But there was a clear gap: there was no indication of what we should expect in our particular setting because the rigorous precision of the models hamstrung their generality. We initially thought that our failure to understand was our own, but when the Anthropic paper was released, it became clear that the failure was the field’s.
>
> To drive the point home, we can use the recommended Arxiv link. Page 4-5 states that we can consider a linearization so long as Equation 1.3.8. holds:
>
> $L_n \leq \frac{1}{||\Phi^+ (y - f_n(\theta_0) ||_2 \phantom{\cdot } ||\Phi^+ || }$
>
> When will this hold? When should one expect this to be applicable to their setting? The first major result (Theorem 1 on Page 11), while extremely precise, reveals zero intuition. Moreover, the models considered assume additive noise, which erroneously leads the reader to believe that noise is necessary, when in fact, noise is not necessary.
>
> > It's unclear how this advances understanding beyond existing literature.
>
> Thank you for giving us a chance to clarify. These are our contributions:
>
> - To the best of our knowledge, our paper is the first to enumerate the three factors that cause the divergence, or explain their significance and how they interact. Many have discussed the small non-zero singular values, but this is only one of three factors and is insufficient by itself. Perhaps you know of a reference explicitly and prominently identifying the three factors?
> - Our paper identifies these factors in the simplest model possible, without making hyperspecific and limiting assumptions, while still describing the phenomenon of interest in its entirety
> - Our paper clarifies several misconceptions, including: (a) the divergence does not require deep networks or nonlinear models; (b) the divergence does not require noise and can occur in a fully deterministic setting, a point that was previously obscured because multiple papers require assuming noise; (c) The divergence does not have a single cause and in fact requires all three factors to be present in order to occur.
> - Our paper also provides (approximate) geometric intuition for the Marchenko–Pastur distribution.
>
> If you would like, we would be happy to provide an extended Related Work section detailing what models and assumptions key previous papers considered. Perhaps this would be a satisfactory way to explain how our paper differs from theirs?
>
> > I think this analysis is a nice way of thinking about double descent, like for a blog post or a lecture note, but I do not see any significant contribution.
>
> We adopted a methodology analogous to the model-experimental systems approach, prevalent in the natural sciences, in which simple and controllable toy models are studied. The goal is to develop an intuitive yet quantitative understanding of deep learning phenomena. Multiple published papers have built and non-rigorously analyze models of deep learning phenomena, with a prominent recent example of grokking [1, 2, 3, 4].
>
> [1] Power et al. Grokking: Generalization Beyond Overfitting on Small Algorithmic Datasets.
>
> [2] Liu et al. Towards Understanding Grokking: An Effective Theory of Representation Learning.
>
> [3] Thilak et al. The Slingshot Mechanism: An Empirical Study of Adaptive Optimizers and the Grokking Phenomenon.
>
> [4] Nanda et al. Progress measures for grokking via mechanistic interpretability.
>
>
> We hope this response clarifies the novelty and significance of our research. We are committed to further refining our paper to address these concerns and look forward to the opportunity to improve our manuscript based on your valuable feedback.

---

> ### Author Response · Authors · 2023-11-22
> **Response to Reviewer 7Ltc**
>
> Based on our reviews, this paper will likely be rejected and it doesn't make sense to invest additional time (or ask more of yours). We think it makes more sense to submit to the ICLR 2024 blog track https://iclr-blogposts.github.io/2024/about/. Thank you for your feedback and time.

---

### Official Review · Reviewer_yGEF · 2023-11-07

**Soundness:** 2 fair
**Presentation:** 2 fair
**Contribution:** 2 fair
**Rating:** 1
**Confidence:** 4

**Summary:**

This paper studies the double descent phenomenon in linear regression at the interpolation threshold. The authors decompose the test error into 3 components: 1. modelling (or irreducible) error, 2. bias and 3. variance.

The analysis provided is based on elementary linear algebraic arguments.

**Strengths:**

The problem considered is of broad interest in machine learning theory.

**Weaknesses:**

The main question is not formed well enough, as a result the main message is described only at a high level. This is unsatisfactory from a technical point of view.

The authors set out with a goal to understand why test loss diverges at the interpolation threshold, but the analysis does not provide a crisp enough answer to warrant looking at the simplified model of OLS.

The arguments are not statistically precise, which makes them hard to trust.

The points where the discussion is high level, e.g. the geometric intuition, the writing is hard to follow and the authors do not get their point across satisfactorily.

The discussion around prior work is very limited and does not fully and clearly explain what was known previously and what gap is this filling in the existing literature.

**Questions:**

The plots contain number of samples on the x-axis which leads to an inverted double descent curve (overparameterized models on the left of the threshold, underparameterized on the right). It might be better to plot the same with 1/n on the x axis, as is common in prior works, or state this different choice very clearly and justify it. The x-labels, y-labels, and plot titles, are too tiny and unreadable, please increase them.

The models considered are regularized (singular value thresholding). This is somewhat equivalent to ridge regression, which we know does not have the double descent if tuned optimally (Nakkiran et al. 2020, Optimal Regularization Can Mitigate Double Descent). As a result, many of the models considered in the overparameterized regime do not interpolate.

---

> ### Author Response · Authors · 2023-11-18
> **Response to Reviewer yGEF**
>
> We sincerely appreciate the time you have invested in evaluating our manuscript. We are eager to address your concerns to enhance the quality of our work.
>
> ## Line-by-Line Response
>
> > The models considered are regularized (singular value thresholding) [...]
>
> We would like to clarify a factual inaccuracy regarding the use of singular value thresholding in our paper. Neither the mathematical model (OLS)  nor Figures 2, 4, 5, 6, 7 have singular value thresholding. Only Figure 3 has singular value thresholding, specifically to make the point of singular value thresholding.
>
> > As a result, many of the models considered in the overparameterized regime do not interpolate.
>
> We would also like to clarify this factual inaccuracy. The mathematical model (OLS) does interpolate in the overparameterized regime, and Figures 2, 3, 4, 5 likewise show interpolation in the overparameterized regime.
>
> > The arguments are not statistically precise, which makes them hard to trust.
>
> We appreciate your point on the utility for statistical precision in arguments. However, our perspective is that many papers have already presented such precise arguments, but require overly specific and complex assumptions e.g. (8+m)-moments, that obscure core concepts and lead to incorrect beliefs such as: (1) small non-zero singular values are the sole cause of the divergence, or (2) noise is necessary for the divergence.
>
> Our paper seeks to clarify such misunderstandings.
>
> >  the analysis does not provide a crisp enough answer to warrant looking at the simplified model of OLS.
>
> As we state in the paper, OLS is the simplest possible model we know of that exhibits the phenomenon of interest and that has known closed-form solutions which can be analyzed and understood; additionally, by using the closed-form solutions, we can exclude the choice of learning algorithm and its (possible) learning dynamics. We could analyze a more complicated model, but doing so would require significantly more effort and we doubt it would contribute much more.
>
> Can you please clarify what you find inadequate about this rationale?
>
> > The discussion around prior work is very limited and does not fully and clearly explain what was known previously and what gap is this filling in the existing literature.
>
> Our paper states what gap it fills in the literature:
>
> - Abstract: “analytically solvable models in this area employ a range of assumptions and use complex techniques from random matrix theory, statistical mechanics, and kernel methods, making it difficult to assess when and why test error might diverge”
> - Introduction: “a comprehensive understanding of why test loss behaves erratically at this threshold remains elusive. Many analytical models aiming to explain this behavior rely on a plethora of assumptions (e.g., i.i.d additive Gaussian noise, sub-Gaussian covariates, (8 + m)- moments) and use advanced proof techniques from random matrix theory, statistical mechanics, and kernel methods. This complexity muddies the waters, making it challenging to pinpoint the precise conditions leading to test error divergence”
>
> To the best of our knowledge, we know of no other paper that identifies the three interpretable factors that we do. Many have discussed small non-zero singular values, but not the other two interpretable factors.
>
> Would the reviewer recommend an extended “Related Work” section to be added, explicitly stating what previous work showed and how our work differs?
>
> > The points where the discussion is high level, e.g. the geometric intuition, the writing is hard to follow and the authors do not get their point across satisfactorily.
>
> Regarding the clarity of our high-level discussions, such as the geometric intuition, we seek your specific feedback to improve our exposition. If there are particular sections or paragraphs, like “Divergence at the Interpolation Threshold”, that require more clarity, please let us know. We are committed to revising these sections to ensure they are accessible and clear to our readers.
>
>
> ## Overall Response
>
> We would like to again express our gratitude for your time and effort in reviewing our manuscript. Your insights are invaluable to the refinement of our work. We have carefully considered your comments and would like to address two areas where we believe further clarification could enhance our mutual understanding.
>
> 1. **Regarding Factual Accuracies:** We have noted a few instances in your review where the interpretation of our work might not align with what was presented in the manuscript. We would be grateful if you could revisit these sections, as we are keen to ensure that our work is evaluated based on its actual content.
>
> 2. **On Specificity in the Review:** We also observed that some aspects of the review discuss our paper in general terms, raising concerns that we address in the main text. It would greatly assist us if you could respond to the text in a specific, detailed manner.

---

> > ### Comment · Reviewer_yGEF · 2023-11-21
> > **Plot x axis with p/n**
> >
> > The reason the double descent curve is called as such is because there previously existed a U-shaped curve in Statistics and ML, which also used number of parameters as the x axis.
> >
> > Please plot all your plots with the x axis proportional to the number of parameters. Otherwise the contributions of this paper are hard to interpret for an average reader who is familiar with the double-descent phenomenon.
> >
> > I will respond in more detail/specificity a little later.

---

> > > ### Comment · Reviewer_5RFc · 2023-11-21
> > >
> > > _Please note this comment is coming from another reviewer, not the authors_.
> > >
> > > Dear Reviewer yGEF,
> > >
> > > I just wanted to add a brief comment to the point made in this comment. While I agree that double descent is typically measured using the number of parameters on the x-axis, it is not invalid to use the number of samples as an alternative as has been pointed out in previous works (see [1,2]). Choosing between these options is effectively arbitrary in my view as they express the same phenomenon.
> > >
> > > That said, I agree that it would be beneficial for the authors to _either_ (a) clearly mention in the text that plotting the number of samples on the x-axis results in the overparameterized regime falling on the left rather than the right **or** (b) remake the plots with $\frac{1}{N}$ on the x-axis such that this aspect is consistent with works that plot parameters on the x-axis.
> > >
> > > Best wishes,
> > >
> > > Reviewer 5RFc
> > >
> > > [1] Nakkiran, Preetum. "More data can hurt for linear regression: Sample-wise double descent." arXiv preprint arXiv:1912.07242 (2019).
> > >
> > > [2] Nakkiran, Preetum, et al. "Deep double descent: Where bigger models and more data hurt." Journal of Statistical Mechanics: Theory and Experiment 2021.12 (2021): 124003.

---

> ### Author Response · Authors · 2023-11-22
> **Response to Reviewer yGEF**
>
> Based on our reviews, this paper will likely be rejected and it doesn't make sense to invest additional time (or ask more of yours). We think it makes more sense to submit to the ICLR 2024 blog track https://iclr-blogposts.github.io/2024/about/.

---

### Author Response · Authors · 2023-11-20
**Key Changes to New Manuscript Uploaded on Nov 19th, 2023**

To all reviewers: We rewrote the manuscript and added new results. The key changes are:

1. Tightened up the mathematical notation and language
2. Improved the verbal geometric intuition for the Marchenko–Pastur distribution
3. Added a new section on constructing adversarial training data. Specifically, we use our insights to design adversarial training data, by which we mean training data that results in a learned linear model with the same training loss but 1-3 orders of magnitude higher test data. The key insight is that the test error of the \textit{learned model} is increased due to errors that the \textit{best possible model in the model class} makes on the training data.
4. Replace figures' PNGs with PDFs for increased readability
5. Moved Figured 1 towards end of paper
6. Added additional citations requested by reviewers

We will post an updated manuscript tomorrow containing more changes suggested by Reviewers 5RFc and iMad.